# What Does “the RNA World” Mean to “the Origin of Life”?

**DOI:** 10.3390/life7040049

**Published:** 2017-11-29

**Authors:** Wentao Ma

**Affiliations:** Hubei Key Laboratory of Cell Homeostasis, College of Life Sciences, Wuhan University, Wuhan 430072, China; mwt@whu.edu.cn

**Keywords:** the essence of life, Darwinian evolution, the origin of evolution, self-sustaining, protocell

## Abstract

Corresponding to life’s two distinct aspects: Darwinian evolution and self-sustainment, the origin of life should also split into two issues: the origin of Darwinian evolution and the arising of self-sustainment. Because the “self-sustainment” we concern about life should be the self-sustainment of a relevant system that is “defined” by its genetic information, the self-sustainment could not have arisen before the origin of Darwinian evolution, which was just marked by the emergence of genetic information. The logic behind the idea of the RNA world is not as tenable as it has been believed. That is, genetic molecules and functional molecules, even though not being the same material, could have emerged together in the beginning and launched the evolution—provided that the genetic molecules can “simply” code the functional molecules. However, due to these or those reasons, alternative scenarios are generally much less convincing than the RNA world. In particular, when considering the accumulating experimental evidence that is supporting a de novo origin of the RNA world, it seems now quite reasonable to believe that such a world may have just stood at the very beginning of life on the Earth. Therewith, we acquire a concrete scenario for our attempts to appreciate those fundamental issues that are involved in the origin of life. In the light of those possible scenes included in this scenario, Darwinian evolution may have originated at the molecular level, realized upon a functional RNA. When two or more functional RNAs emerged, for their efficient cooperation, there should have been a selective pressure for the emergence of protocells. But it was not until the appearance of the “unitary-protocell”, which had all of its RNA genes linked into a chromosome, that Darwinian evolution made its full step towards the cellular level—no longer severely constrained by the low-grade evolution at the molecular level. Self-sustainment did not make sense before protocells emerged. The selection pressure that was favoring the exploration of more and more fundamental raw materials resulted in an evolutionary tendency of life to become more and more self-sustained. New functions for the entities to adapt to environments, including those that are involved in the self-sustainment per se, would bring new burdens to the self-sustainment—the advantage of these functions must overweigh the corresponding disadvantage.

## 1. What Does “the Origin of Life” Mean?

To avoid discussing this issue in an ambiguous context, we ought to clarify what on earth is “life”. Unfortunately, in respect of the latter question, few of us can offer a convincing answer [1,2]. Indeed, the situation is so awkward that we cannot even present a clear definition of life in any textbooks of biology, although the field has gained so many great achievements to date.

Recently, I made an assertion about the reason underlying this situation [3]. That is, for the concept of life there are two completely different aspects, Darwinian evolution and self-sustainment, which are usually talked about together. For example, a famous working definition from NASA, which are aiming at the exploration for extraterrestrial life, says: “Life is a self-sustaining chemical system capable of undergoing Darwinian evolution”. How can a chemical system, as an entity, undergo Darwinian evolution? Darwinian evolution means the change of the form of life over generations, not the change of a certain entity per se. Thus, it would be better to split the definition of life, for example, by adopting some expression like: “A life form is a matter form capable of undergoing Darwinian evolution; a living entity is a self-sustaining chemical system—in nature, it results from the Darwinian evolution and might engage into further Darwinian evolution” [3].

So, we know that the origin of life should also imply two distinct issues: the origin of the life form—or say, the origin of Darwinian evolution, and the origin of living entities—or say, the origin of the self-sustainment. Then comes a question of “order”: which originated at the very beginning? In fact, there has long been a disputation concerning “replication first or metabolism first” during the emergence of life [4,5,6,7]. This disputation, if brought closer to the two essential aspects of life, should have largely represented the “order question” here, i.e., “Darwinian evolution first or self-sustainment first” (but note that replication, obviously, is not enough for Darwinian evolution, and metabolism is not all that the “self-sustainment” means).

To answer the “order question”, we should examine the relevant concepts more carefully. Actually, as it was noted, “self-sustainment” has a rather blurry meaning [3]. What does the wording “self-” mean? Indeed, we may say that it means the entity synthesizes its own components in an active way, functionally depending upon its own components. However, the problem is: what on earth are “its own components”? For example, the idea of “metabolism first” often assumed that some autocatalytic cycle (or something alike) may have appeared before the emergence of genetic molecules [8,9]. It seems that such a reaction cycle can be seen as a “self-sustaining” chemical system. The reactants involved within the cycle may constitute its own components. Indeed, here, the autocatalytic system, like a machine, produces its own components. However, the really important thing is what would occur afterwards. Sooner or later, genetic molecules must have emerged in the system, enabling the Darwinian evolution occurring subsequently, which is undebatable [5]. From then on, the life form would change from generation to generation mainly according to the alteration of the genetic molecules (or say, genes carried on them). Indeed, it may be argued that the reactants involved in the old cycle could be transferred to the next generation, therewith organizing a new cycle and thus still contributing to the self-sustainment of the latter living entity. However, things would turn out to be that genes, by instructing the synthesis of functional molecules (e.g., enzymes), would determine the future metabolic way—likely deviating from or completely deserting the original cycle (accompanying with Darwinian evolution). Additionally, even if the original cycle was maintained, the reactants that were included within this cycle might become easy to synthesize (from other resources) by some enzymes evolving up later, and it would no longer be important to transfer these reactants to the next generation by the cycle itself. So, it was then genes that would ultimately determine what would occur in the living system, and it was the genes that should “manage” to sustain the living system, which could transfer them to the next generation. Obviously, the essence of the machine changed, i.e., the meaning of “self-” became different. 

That is to say, even if the so-called self-sustaining autocatalytic cycle (or something alike) emerged first, the self-sustaining living system emerging after the appearance of genetic materials would have no essential relationship with the original cycle, due to the shifting of the connotation of “self-”. The autocatalytic cycle, if ever existed, seemed to have only made its sense as some environment, providing the necessary energy and raw materials, for the emergence of those early genetic molecules (similar ideas have been expressed previously [10]). Indeed, just thinking about the metabolic system in modern cells, obviously the “own” components should be those enzymes, rather than the reactants (or say, the metabolites). No doubt, genetic molecules and those functional molecules synthesized under the instruction of the genetic molecules are the central components of a living entity (just as we can see in the Central Dogma), thus most approaching the sense of “own” or “self-”. 

To conclude, as to the life’s feature “self-sustainment”, what we really concern about should be the self-sustainment of the system “defined” by the genetic information. Therefore, the problem of the origin of the life can be summarized as: first, the origin of Darwinian evolution, as labeled by the emergence of the genetic material, and next, the arising of the self-sustainment guided by the genes carried on the genetic material. At least, the self-sustainment should not have arisen before the origin of Darwinian evolution, if the possibility for simultaneous advent of the two features could not be ruled out. In terms of the splitting definition of life (mentioned above) [3], that is, living entities should not have appeared before the emergence of the life form.

## 2. What Does “the RNA World” Mean?

### 2.1. About the Idea of the RNA World

As long ago as the 1960s, it was speculated that there might have been an early stage of life in which RNA played both the roles of DNA and proteins [11,12,13]. Indeed, in modern life, DNA directs the synthesis of proteins and proteins catalyze the synthesis of DNA; it seems that both of them are indispensable for the running of life. Then, during the emergence of life, which came first? This became the famous “egg-chicken” paradox in the field of the origin of life. At that time, it had already been known that RNA might act as genetic material (in some viruses). Hence, when it was found that RNA could really act as functional material (i.e., ribozymes) in the early 1980s [14,15], the idea concerning the RNA world took its shape and gained widespread attention in this field [16].

Thereafter, evidence accumulated quickly in favor of the hypothesis [17,18,19]. The most surprising and convincing support came from researches on the ribosome [20,21]. It was revealed that the functional center of the ribosome is composed of RNA rather than proteins! Proteins in the ribosome are peripheral in location and mainly play a role of stabilizing the structure. The ribosome can be seen as a large ribozyme, functioning to synthesize proteins. That is to say, it is very likely that proteins were invented within a world in which RNA, alone, acted as functional materials. The ribosome should have just been a relic dating from the ancient RNA world; or else, why our modern cells do not use proteins to catalyze their own synthesis, given that proteins are obviously better at catalysis?

However, the proponents of the hypothesis almost immediately have to face a further problem: how did the so-called RNA world itself come into being? May it have appeared de novo? In those days, experimental work to simulate the prebiotic synthesis of nucleotides apparently had not transferred messages so optimistic. In fact, this area was even described as the prebiotic chemist’s “nightmare” [22]. Nonetheless, as time went on, advance was achieved regarding the prebiotic synthesis. An eminent study came from Sutherland’s group showing that pyrimidine nucleotides may have been able to be synthesized in large amounts in a way “prebiotically plausible” [23], which had been known as a notoriously difficult problem [22,24]. Later studies along this line suggested that purine nucleotides might also be able to be synthesized using similar strategies [25,26]. The de novo appearance of the RNA world, gradually, became not so inconceivable [27]. Then, should the RNA world have stood at the very beginning of life? In the following part, we will analyze this problem in logic and comment it in light of our current knowledge. Indeed, if we choose to evade this problem, we will never clearly see the significance of the idea of the RNA world.

### 2.2. The RNA World as a Scenario in the Very Beginning of Life

First of all, according to the conclusions that are made in Section 1, the origin of life means first the origin of Darwinian evolution, which is labeled by the emergence of genetic material. Indeed, it is genetic material, as well as the functional material that it encodes (here, by “encoding” I mean “instructing the synthesis of”), that provide the foundation of Darwinian evolution [3]. That is, for any possible scenario concerning the outset of life, there must be genetic material and the functional material it encoded. For the scenario of the RNA world, the solution is simple: RNA can both act as genetic material and functional material. An RNA gene instructed the synthesis of a corresponding ribozyme just via template-directed copying, the same way as it adopted in its replication.

Notably, there could be alternative scenarios adopting a similar solution. For example, in those years, the “nightmare” regarding nucleotides’ prebiotic synthesis was so awful that some people began to search out for other RNA-like polymers, which might also, possibly, act as both functional and genetic material, such as p-RNA (pyranosyl-RNA), PNA (peptide nucleic acid), TNA (threose nucleic acid) and GNA (glycol nucleic acid) [17,22,24,27]. It was supposed that there might have been one (or even more) pre-RNA world(s) before the RNA world. For the so-called pre-RNA world, the solution was similar to that for the RNA world: the RNA-like polymer acted as both genetic material and functional material. One problem with the pre-RNA world idea is the absence of relevant evidence: for example, first, it was assumed, rather arbitrarily, that the RNA-like polymer’s monomers, may have formed more easily in the prebiotic circumstance; next, unlike RNA, we could not find any vestige about the RNA-like polymers in our modern living world. In addition, the scenario lacks a sound justification for the subsequent evolution: why did the transition from the pre-RNA world to the RNA world occur? As a contrast, the RNA world has a sound justification for its later evolution: DNA is in nature better at acting as genetic material (template molecule) [28] due to its greater stability against hydrolysis [29], less proneness to self-folding [30], and higher fidelity in replication [31], and proteins, as it is well-known, are better at acting as functional molecules owing to their residues’ diversity—deriving more abundant chemical activities.

Perhaps somewhat unexpectedly, here it is underlined that there can still be other possible scenarios—adopting a different solution. That is, genetic material and the functional material it encoded, even if distinct from each other, could have emerged together in the very beginning and launched Darwinian evolution as well. This assertion is unexpected because it casts doubt on the logic that is followed by the original proposition of the idea of the RNA world. Indeed, why genetic material and functional material, if different from each other in molecule type, cannot emerge together, especially considering that the “prebiotic pool” may have been a rather complex “soup” [32]? 

For example, if DNA and RNA could both be synthesized in the prebiotic pool, DNA might act as genetic material and encode functional RNA molecules. It should be noted that DNA can simply instruct the synthesis of RNA via template-directed copying, just like the transcription shown in modern cells. The problem with this scenario, however, is that if DNA and RNA were simultaneously synthesized in a common prebiotic environment, the synthetic reactions may have interfered with each other (note: in modern cells, the synthesis of DNA and RNA is strictly controlled, proceeding in different periods or/and different locations). Thereby, mosaic nucleic acids (with a combination of nucleotide and deoxynucleotide residues) may have arisen. A recent interesting study showed that mosaic nucleic acids might also carry functions, albeit less efficient than RNA [33], which suggested the possibility of a mosaic RNA/DNA world at the very beginning—perhaps it was not until later in evolution that pure DNA gradually took the role of genetic material owing to its greater suitability as template and pure RNA took the role of functional material due to its higher efficiency in that aspect. However, it should be admitted that such a scenario is so complicated that a great deal of further exploration, both experimental and theoretical, are needed to evaluate whether it deserves our serious consideration.

Additionally, one may also conceive: if RNA and proteins could both be synthesized in the prebiotic pool, RNA might act as genetic material and proteins may act as functional material. Indeed, there are now some proponents for the hypothesis of the “RNA/proteins (peptides)” world (e.g., [34,35]), especially considering that amino acids (and simple peptides) seem easy to form in the prebiotic environments as people usually believe. However, notably, this version of story is also flawed. Unlike DNA encoding RNA, which can simply be implemented via template-directed copying, there is no straightforward mechanism for RNA to encode proteins. It is here worth emphasizing that the coexistence of genetic material and functional material is not enough, and to kick off the Darwinian evolution, the genetic molecules must, simultaneously, encode the functional molecules. Even though peptides could have been abundant in the environment, they were likely to be only random in sequence and had nothing to do with the sequence of RNAs there around—so, no evolution would happen. Indeed, certain peptide sequences may have preferentially aggregated in certain conditions (e.g., as suggested in [36,37,38,39]). More interestingly, some special peptides, namely oligoarginine, might even benefit RNA replication by facilitating the dissociation of the product chain from the template chain [40]. Nonetheless, things would not be different if the specificity of the peptides were not connected to that of RNA—as genetic molecules. Interestingly, the Direct-RNA-Template (DRT) hypothesis [41], which proposed the correlation of every single amino acids to a segment of RNA, did formulate a direct way for RNA to code peptides. However, the way seems too inefficient to have worked in the very beginning (that is, a fairly large RNA molecule would be required to encode even a short peptide). This mechanism, instead, is more likely to have played its role during the emergence of a “proto-translation machine” in the RNA world, which already existed [42,43,44]. Most recently, it was suggested that the “informatic mapping” between RNA and peptides may have been established from the very beginning due to self-organization [45,46]. The idea is quite fresh, but obviously, a great deal of work is needed to make the relevant scenario clearer.

Taken together, we can conclude that though we have not obtained sufficient evidence to be sure about the existence of an RNA world at the start point of life, and even the logic for the raising of the idea of the RNA world is not, as it is usually taken for granted, so obliged, this world is still most likely to have been the earliest scenario for the life on our planet—at least according to the knowledge that we have so far. 

## 3. What Does “the RNA World” Mean to “the Origin of Life”?

Above we have conceptually inquired into the problem of the origin of life and have interpreted in depth the idea of the RNA world, as a central hypothesis in this field. It is then interesting to relate the two contents to each other and ask: what on earth does “the RNA world” mean to “the origin of life”?

Previously, the process of the origin of life is only pondered and discussed as a blurry and abstract issue. If, as mentioned above, there was an RNA world standing at the start point of life’s history, the origin of this world per se, together with its early development, would have represented the very process of the origin of life. Then, we obtain a “concrete background” to think, discuss, and even debate about this issue. For example, based upon the analysis in Section 1 about the essence of the problem of the origin of life, it is immediately interesting to imagine how the two sub-processes, the origin of Darwinian evolution and the arising of self-sustainment, were manifested in the RNA world. In fact, as it will be shown below, with its relatively simple material-base: mainly RNA—plus amphiphiles, which assembled to form protocells’ membrane, this concrete scenario could greatly promote our understanding on the origin of life. Firstly, let us conceive the concrete scenario by imagining those scenes involved in, in a successive way.

### 3.1. Did the RNA World Start at the Level of Molecule or Cell-Like Vesicle?

Since RNA can both act as genetic molecules and functional molecules, it is natural to think that RNA may have evolved initially just in a naked way, at the molecular level. It is usually supposed that some RNA, catalyzing the template-directed copying of RNA, may have favored its own replication (often referred to as an “RNA replicase”) and spread in a prebiotic pool [17,22,23]. It is popular to assume that this ribozyme emerged first, perhaps mainly because of those early laboratory studies showing the inefficiency of the non-enzymatic template-directed RNA copying [22,24]. However, some recent studies began to demonstrate more efficient non-enzymatic copying of DNA/RNA in these or those conditions [47,48,49,50]. That is to say, other functional RNAs, which may have favoring their own replication, such as a ribozyme catalyzing the synthesis of the building blocks (i.e., a nucleotide synthetase ribozyme) [51], should also be considered as candidates of the functional RNA emerging first [52].

To avoid a diverging representation, here let us adhere to the assumption that the replicase emerged first. A key problem of the scene regarding the emergence of the replicase is the parasite problem; that is, since other RNA sequences might also exploit the ribozyme, how could the replicase “win the game”? Through theoretical work in this area, it is now well known that the spatial limitation may have played an important role for the replicase to overcome the parasite problem [53,54,55,56]. As to the actual environments, for example, it was imagined that mineral surfaces [57,58,59], lacunose ices [60,61,62], or porous rocks [63,64,65] may have provided the spatial limitation and become the hatchery of early life on the earth. On the other hand, it was proposed that “tag mechanism” (i.e., the replicase recognizing its target templates through a short subsequence on the templates) might have served as a strategy for an RNA replicase to resist parasites [22,66]. A suspicion concerning this strategy seems to be that parasites could be equipped with a tag as well [67]. A recent computer simulation work in our group revealed that the tag mechanism could indeed take effect and suggested that it may have worked as an important complement to the mechanism of spatial limitation for the replicase to resist parasites [68]. As it turned out, somewhat surprisingly, but well as a solution to that suspicion, the reason why the tag mechanism can work is not that it favors the replicase directly but that it suppresses the appearance of parasites—the requirement of including the tag could have seriously restrained the de novo arising of parasites (for details, please see the original work [68]).

Certainly, if the RNA molecules were encompassed within a lipid vesicle, forming a so-called protocell, then the parasite problem would be much less serious. For example, it was suggested—not surprisingly—that the simplest protocell may have been an RNA replicase trapped in a lipid vesicle [69]. Parasites outside would be kept away by the membrane. Parasites appearing inside the protocell, which should be mainly derived from degradation or partial replication of the RNA replicase, would constitute the main problem. The tag mechanism might aid in suppressing the appearance of these parasites. If only a few parasites arose therein, they might be got rid of with the protocell’s division—by chance. More importantly, even if the inner parasites ultimately “ruined” a protocell, that is, the RNA replicase disappeared in the vesicle, other protocells would not be influenced by these parasites—unless eventually the parasites evolved the capability to invade other protocells, like viruses in our modern living world. Interestingly, a recent experimental study using a simple RNA replication system, though only “translation-coupled”, demonstrated such effects of cell-like compartment against parasites in the real world [70].

A logic justifying the idea of “a naked stage first” is: “the simpler, the more likely to emerge de novo”. However, if the prebiotic pool was a rather complex soup [32], for example, comprising those lipid molecules, as well as the blocks of RNA, then this logic would fade out. In fact, the key problem about the scene of a protocell trapping an RNA replicase is not related to this logic, but concerns the permeability of the building blocks of RNA. If nucleotides are difficult to access, how can the RNA replication within the protocell keep going on? Indeed, it was argued that the prebiotic membrane—e.g., composed of fatty acids (instead of phospholipids like modern cells)—may have been more permeable to nucleotides [71,72]. However, such a membrane is rather unstable in a circumstance with high concentration of Mg^2+^, which is a usual condition for RNA’s template-directed copying [73,74]. Though there may be some remedial measure, e.g., by shielding a portion of Mg^2+^’s surface with citric acid, obviously, the scene is not quite convincing as the authors admitted in the same paper [75]. Interestingly, one might assume that a protocell containing a nucleotide synthetase ribozyme (instead of an RNA replicase) emerged first. Surely, if what needed to permeate into the vesicle are only the precursors of nucleotides, things would be quite different. For instance, it has been proved by experiments that it is much easier for ribose than for nucleotide to permeate across a lipid membrane, no matter what the membrane is composed of, fatty acids or phospholipids [72,76]. Indeed, if non-enzymatic template-directed RNA copying is sufficiently efficient, as implied in some recent experiments [47,48,49,50] (mentioned above already), then this alternative idea deserves serious consideration.

### 3.2. From Molecular Form to Cellular Form

No matter how, let us come back to our mainline as it is assumed: an RNA replicase emerged first in a naked way. Indeed, on account of the belief about the importance of this functional RNA to the origin of the RNA world, there have been long-standing efforts to construct such a ribozyme in laboratory [77,78,79,80,81,82]. Up to date, by in vitro evolution, an RNA polymerase ribozyme that is capable of copying some RNA templates as long as itself (about 200 nt) has been acquired [82]. Wherein, base-pairing between the tails of the ribozyme and the RNA template contributed a lot to the efficiency-improvement of the polymerase (the significance of such inter-tethering had also been suggested in previous studies [77,80]). However, the target RNA templates are still limited in sequence; namely, the ribozyme cannot yet copy itself or its complementary chain. Additionally, the ribozyme is in itself too long, which leads to two problems: How can such a long RNA, with a definite sequence, emerge de novo from the prebiotic pool? How can the long double chain, resulting from one turn of copying, dissolve, thus allowing for a next turn of copying? (Oligoarginine peptides might have some effects as mentioned above [40], but it is doubtful that they can work to an extent involving such long RNA molecules).

Therefore, the search for shorter RNA replicase, even with some sacrifice of efficiency, would be an urgent task for supporting this “earliest scene” of the RNA world. Now that the tail-binding of the ribozyme onto the template can greatly improve the ribozyme’s efficiency, it would be natural to expect that a shorter version of the replicase (perhaps about 30–50 nt long), with the tail-binding retained, may have had an efficiency, albeit somewhat lower, yet enough to support its own thriving in the system. Importantly, the tail subsequence on the target template, which is recognized by the replicase through base-pairing can just serve as a so-called “tag”. As mentioned above, one of our theoretic studies on evolutionary dynamics (by computer simulation) has concluded that the introduction of a tag mechanism would greatly enhance the replicase’s ability to resist parasites [68]. Maybe this would turn out to be a paradigm for the assertion that a conclusion from one aspect of the problem of life’s origin may aid the exploration of another aspect (please see [83] for a discussion about the three aspects of the origin of life: evolutionary, chemistry, and history; here I mean the evolutionary study concerning the tag mechanism directs the research on the chemical exploration of the RNA replicase).

Let us go on with our scenario. What may have occurred after the emergence of the RNA replicase? Indeed, even if a nucleotide synthetase ribozyme had not emerged before the emergence of the replicase, here it seems inevitable to have emerged subsequently. When the replicase spread in the pool, nucleotides would become scarce, and a ribozyme that could supply the building blocks would be strongly favored in the context of natural selection. The nucleotide synthetase ribozyme would enable the system to exploit more fundamental materials, namely precursors of nucleotides. It has been demonstrated by computer simulation from our group [84] and Higgs’ group [85] that an RNA replicase and a nucleotide synthetase ribozyme can cooperate in a naked scene. If so, it is conceivable that it was then the nucleotide precursors’ turn to become scarce. Thus, there may be selective pressure for the emergence of ribozymes to exploit materials that are further fundamental, namely precursors of the nucleotide precursors. However, the cooperation of three or more ribozymes, without an explicit boundary in space, would become difficult. Indeed, as it was noted in the paper from Higgs’ group, “the sensitivity of this two-ribozyme system suggests that evolution of a system of many types of ribozymes would be difficult in a purely spatial model with unlinked genes” [85]. 

That is, the subsequent scene (before the emergence of the third ribozyme) is likely to have been the appearance of protocells. In addition, it is worth noting that, after the emergence of the nucleotide synthetase ribozyme, the appearance of protocells would have become more feasible (as mentioned above)—what need to permeate into the vesicle are, instead of nucleotides, only precursors of nucleotides. The two ribozymes, if by chance, were “engulfed” into some lipid vesicle, would then cooperate within a protocell. 

### 3.3. From Pseudo-Protocell to True-Protocell, then to Unitary-Protocell

Initially, the protocell should have yet been unable to synthesize the components of its membrane. The membrane appears to have been no much more than an environmental factor. We would like to call it a “pseudo-protocell” [84]. Interestingly, it was suggested by an experimental work that the membrane of such pseudo-protocells may have been able to grow simply on account of the growth of their “contents” [86]. The protocell, as it was explained, would become swollen on account of osmotic pressure when the RNA molecules therein increase in quantity due to their replication; then, the lipid molecules on the membrane would be “unwilling” to leave the membrane. If the rate for lipid molecules in the environment to join the membrane is constant, the vesicle would become larger increasingly. On the contrary, those protocells with a lower rate of content growth would shrink due to the same interchange of lipid molecules between the membrane and the environment. When a protocell becomes larger and larger, it would become more and more unstable on account of the physical mechanisms that are involved in the membrane’s self-organization. Then, it may tend to divide into two or more smaller ones due to certain perturbation coming from the circumstance [71,87], thus completing a circle of reproduction (note that at cellular level, “reproduction” is a more proper word than “replication” [55,88]). Taken together, as it seems, a competition of RNAs’ replication at the molecular level would result in a reproduction competition at the cellular level. In this way, therefore, even with their membrane independent of any genetic features, the pseudo-protocells could have shown their feature at a cellular level. 

Nonetheless, at that time, there should have been a strong selection pressure for the emergence of a ribozyme to synthesize the lipid molecules. This ribozyme would provide the membrane components from the inside and allow or the protocell’s membrane to grow faster, thus achieving the potential to accommodate more contents. Then, the replication of the inner RNAs would be promoted when more raw materials could flow in. In contrast to the situation for pseudo-protocells, which is the content-triggered growth, here is a sort of membrane-triggered growth. This growth, of course, would also eventually lead to “cell division” due to the instability of the protocells, thus completing the circle of reproduction. Indeed, in one of our modeling studies, the emergence of such a ribozyme has been demonstrated to be favored in evolution [89] because protocells with this ribozyme would reproduce faster than those without it. We call such a protocell, which has the capability to synthesize its membrane components, thus engaging in the membrane growth of its own, a “true-protocell” [84].

Up to this stage, though lipid vesicles provided a good mechanism for the cooperation of different ribozymes, accompany with the emergence of more ribozymes, another problem would become increasingly apparent. That is, due to the random distribution of the ribozymes between offspring protocells during the “cell division”, there is a risk of “gene loss” [55,90]. Obviously, the more the genes (i.e., the ribozymes), the more probable one or more genes would be absent in an offspring protocell after the division. As it was pointed out, there should have been a selective pressure for a strategy of linking the genes together, forming a “chromosome” [90]. However, there are two main problems with this strategy. First, the chromosome would be much longer than single genes, how could it sustain for a sufficient long time against degradation (especially when considering that RNA is rather fragile in chemistry)? Second, how could the ribozymes, at this primitive stage, be transcribed from the genes locating on the chromosome? We noticed that it had been proposed that viroids in modern life world might have been a relic of the RNA world [91]. Inspired by the mechanism involved in the replication of viroids, we speculated that a circular chromosome with self-cleaving elements (e.g., the hammer head ribozyme) spacing the genes thereon may have solved these problems: by adopting a circular form, while the intra-chain degradation is still at risk, the degradation at chain ends, a more serious issue, could be avoided; by introducing the small self-cleaving elements, the plus chain of the RNA chromosome could easily break into separate ribozymes. We showed by computer simulation that this strategy is effective and may lead to the prosperity of the protocells containing such chromosomes [82]. Certainly, since tag mechanism may have been introduced into the RNA world quite early (as mentioned above [68]), it is also conceivable that there is an alternative for the self-cleaving mechanism: if every gene had a tag to label its start point on the chromosome, it may have been easy to “read out” (“transcribed” by a ribozyme identical or similar to the replicase). No matter how, we name such a protocell, which has all its genetic features borne on a chromosome, a “unitary-protocell” [84]. The emergence of unitary-protocells may have been rather important in the history of life’s evolution—from then on, the emergence of more functions may “simply be realized” by introducing more genes into the chromosome [92].

In fact, quite a few scenes above concerning the origin and development of the RNA world have been described previously [83,93]. What I really want to discuss here is how these detail scenes could make a concrete story in regards to the origin of life, which was previously only an abstract issue for all of us. See below for details.

### 3.4. About the Origin of Darwinian Evolution

As mentioned above, the origin of life is, firstly, the origin of Darwinian evolution. Then, how may Darwinian evolution have got started? According to the scenario described above, the RNA world may have begun at the molecular level. From the prebiotic pool, nucleotides may have been synthesized abiotically, and linked together to form RNA—perhaps by mineral catalysis [58,93]. Then the pool should have become full of random RNA species, with different lengths and diverse sequences. In an “elegant” version of the story, these random RNA species were just the first Darwinian entities (entities with a form capable of undergoing Darwinian evolution [3]): they replicated via non-enzymatic-template-directed copying and competing for raw materials. When by chance certain RNA species with a function favoring RNA replication appeared in the pool—by mutation (in replication) or random ligation/recombination of the pool RNAs, it could have spread in the system (provided that it operated above Eigen’s error threshold for its own replication). The emergence of the functional RNA should have represented the first step of Darwinian evolution. If so, according to the mainline in our story, the functional RNA in the spotlight of this scene was just the RNA replicase. However, things may also turn out to have not been so elegant. If non-enzymatic-template-directed copying was then rather inefficient, the RNA replicase itself may have been the first Darwinian entity, provided that it could appear de novo via random polymerization of nucleotides or by random ligation/recombination of the pool RNAs. That is, Darwinian evolution may have been able to begin only after the emergence of the replicase. The initial Darwinian evolution may have been represented by the emergence of more efficient replicases thereafter.

No matter how, then, a nucleotide synthetase ribozyme is likely to have emerged and become co-thriving with the RNA replicase in the naked stage. Indeed, Darwinian evolution in this scene may have taken some sense of a higher level—in regard to the locally distributed clusters of the two ribozymes [85]. However, without a clear boundary, the benefit arising from one cluster would have easily been exploited by other clusters, and there was not a definite unit for the higher level evolution. That is, Darwinian evolution here, basically, remained going on at the molecular level.

It is only when the two ribozymes were encompassed within lipid membranes that Darwinian evolution at a higher level began to show clearly. The fate of the functional RNAs would be tightly tied up with the fate of the protocell they resided in. For pseudo-protocells, on account of the “osmotic pressure” effect, the more efficient replication of functional RNAs would directly result in a faster reproduction of the protocells containing them. Even a functional RNA having no direct influence on the synthesis of RNA may have been favored in the evolution as long as it favored the reproduction of the protocell. As a case in point, the protocells containing a ribozyme favoring the synthesis of membrane components would reproduce faster than those without the ribozyme [84,89]—so this ribozyme could emerge, marking the emergence of true-protocells in the RNA world.

The emergence of the unitary-protocell, that is, a protocell with all its genes linked into a chromosome is significant for the Darwinian evolution at the cellular level. It has been pointed out that even when the protocells appeared, the competition at the molecular level went on. That is, at the stage of protocell, there is still a two level Darwinian evolution: the molecular level and the Darwinian level [55,94,95]. Actually, even in our living world, multilevel evolution is inevitable, as vividly shown in the famous scientific monograph “Selfish Gene” by Richard Dawkins [96]. Indeed, in the case here, functional RNAs within a protocell, for example, an RNA replicase and a nucleotide synthetase ribozyme, would, beyond their cooperation, compete against each other because they have the same building blocks. It may be critical to balance their relative quantity to achieve a better development of the protocell. With the increase of the types of the cooperative ribozymes, the balance would be more subtle. However, when the genes were linked into a chromosome—replicated and passed on together to the offspring, the detriment that is associated with the competition at the molecular level would be greatly alleviated, and Darwinian evolution would then run largely at the cellular level, being more efficient. In fact, it is even not inappropriate to assert that Darwinian evolution at a cellular level took its genuine start only when the unitary-protocell appeared. The chromosome, as an RNA molecule itself, served as the ultimate target of natural selection. Indeed, one molecule is sufficient to carry multiple genotypes of a complex life form because genes that are aligned on the molecule can be replicated the same way one after another. However, due to the golden rule “one molecule, one function” (e.g., one enzyme or one ribozyme, typically, could only catalyze one reaction), a single molecule seems far from being sufficient to bear multiple phenotypes of a complex life form. That is just why a ribozyme, e.g., the RNA replicase, at the very beginning, in itself, cannot evolve too far ahead. The chromosome, however, evaded that golden rule by resorting to the ribozymes transcribed from it (instead of itself) to carry the functions. Certainly, this is impossible until the emergence of protocells, whose lipid membrane acted to hold all of the transcripts (i.e., ribozymes) together there around. Indeed, from then on, the emergence of more functions could simply be realized by introducing more genes into the chromosome. In other words, it is in this way that our living world achieved its fundamental potential to become complicated.

### 3.5. About the Arising of Self-Sustainment

The arising of self-sustainment is another issue for the origin of life. If Darwinian evolution started at the molecular level, e.g., as described in the scenario we focused on—an RNA replicase emerging first in a naked way, there would have been no problem of self-sustainment at that time. There is no sense for a molecule to self-sustain. The key reason is that the covalent bonds within a molecule would prevent it from refreshing—a molecule either stays unchanged or is “ruined” on account of any change (typically, degradation). If self-sustainment is a key feature for a living entity, the replicase molecule is not “alive” (like viruses in our modern living world [3]); nonetheless, it can engender “offspring” (by replication), and thereby engage into Darwinian evolution. This is easy to appreciate.

Then, when a nucleotide synthetase ribozyme ensued, what about the problem of self-sustainment? Indeed, the emergence of the nucleotide synthetase ribozyme would have rendered the whole system more self-sustaining, by exploiting more fundamental raw material. However, without a clear boundary, there would be no definite entities at a level above the level of molecules. Therefore, self-sustainment still made little sense at this stage.

When the two ribozymes were encompassed into the lipid membrane and formed protocells (pseudo-protocells), things became different. The RNA molecules within the vesicle could be replenished if some of them were degraded, and the membrane components (lipid molecules) could also be replenished if some of them degraded or left the membrane. Here, it becomes clear that the key factor rendering “self-sustainment” meaningful is the establishment of entities at a level above that of molecules.

For a protocell containing the two ribozymes, it should be noted that, while these functional RNA molecules may be replenished via replication, the lipid molecules on the membrane could only be replenished by recruiting new ones from environments. The self-sustainment of such protocells was apparently “incomplete”. That is just why we named them “pseudo-protocells” [84]. When an amphiphilic molecule synthetase ribozyme emerged, the RNA-based protocells would participate in the synthesis of their own membrane components, and thus were obviously more self-sustained—therefore named “true-protocells” [84] (notably, this is a typical case showing that the meaning of “self-” is tightly connected with genetic features and their corresponding functions). Certainly, more ribozymes, like some nucleotide precursor synthetase ribozymes and some amphiphile precursor synthetase ribozymes may have emerged subsequently and have rendered the protocells more and more self-sustained. But as mentioned above, before the emergence of a number of genes, a chromosome, which linked the genes together, and thus avoided gene loss during the random division of the protocell, should have emerged. The unitary protocells, containing the chromosome, would progress to advanced forms—forming complex metabolism pathways, gaining more controls upon its own growth and division, and, ultimately, of course, evolving into the DNA/proteins living world.

Here, it is worth talking about the relation of replication, self-sustainment, and reproduction. It has been noted that there is a difference between replication and reproduction [55,88]. Indeed, this is a helpful identification in concepts. After the emergence of a protocell, it is reproduction, instead of replication, which would meet the basic requirement of Darwinian evolution in the respect of generating offspring. The reproduction of an RNA-based protocell means the replication of the RNA molecules inside the vesicle, the growth of the membrane, and then the division of the protocell [55,88]. In the stage of protocells, obviously there is a large portion of overlap between the mechanism of self-sustainment and that of reproduction. When the replenishment of RNA molecules and membrane components exceeds the need of self-sustainment, the protocell would grow, and finally divide, resulting in reproduction. However, when life became more advanced and was able to take more controls on its own self-sustainment, growth, and reproduction, the processes would then become more and more distinguishable from one another. In light of the situation of our modern living world, we can understand the “final” scene as: for a living entity, while functional factors (phenotypes) contact directly to the environment and are refreshed constantly to support the self-sustainment; the genetic factors (genotypes) are protected well, not refreshed and left for “usage” in the reproduction; the aim of the self-sustainment is to gain a better chance or a higher efficiency for the reproduction, and thus win the corresponding life form of the living entity a superior position in the context of Darwinian evolution. As it relates to the RNA world scenario described above, notably, it is not until the chromosome (thus, the unitary-protocell) appeared that the genotypes and the phenotypes became independent of each other, which made the later emergence of these modern—more specialized and efficient—life forms feasible.

Finally, the tendency to become more self-sustained or evolve other features for adapting to the environment would result in more “components of its own” for the living entity. Thus, the task for its self-sustainment would be “heavier”. The advantage brought about by these new components ought to cover the burden to synthesize them. In a scene mentioned above, for example, for the pseudo-protocells containing an RNA replicase ribozyme and a nucleotide synthetase ribozyme to incorporate a new ribozyme—the amphiphile synthetase ribozyme (which would give rise to the true-protocells), the advantage of synthesizing their own membrane components must outweigh the burden in respect to producing such a ribozyme. Likewise, the task per se should not be too heavy to synthesize those new “metabolic” ribozymes, with which the protocells could go further to exploit raw materials that are more fundamental than the nucleotide precursors and amphiphile precursors. Certainly, a most “typical” case in this point should be the one concerning the emergence of proteins in the RNA world. That is, the burden in respect to sustaining the translation machine (though rather complex) should have been trivial when considering the great advantage of the introduction of proteins, as a significantly more efficient functional material.

### 3.6. Comments

Here, we have talked about the origin of Darwinian evolution and the arising of self-sustainment in a concrete background of the RNA world scenario. As it is implied in the text, though we say that the RNA world is most likely the earliest stage for life, this assertion is far from certain, especially considering that the idea of the RNA world is not obliged in logic and our knowledge on prebiotic environments and prebiotic chemistry remains quite limited. However, at least, we can benefit a lot from the way to talk about the origin of life—previously a rather abstract problem—based upon a concrete scenario. In particular, the RNA world scenario has a simple base of material, that is, the central molecules, genetic and functional, are identical in material. Such a “succinct” world takes our focus to the core issues of the problem: the origin of Darwinian evolution and the arising of self-sustainment. If the earliest scenario in the history of life is, in the future, shown likely to be a different one, either a different version of the RNA world scenario or even a non-RNA-world story, the discussion here may still be helpful for the new interpretation possibly involved because the principles concerning the implementation of Darwinian evolution and that of self-sustainment remains unchanged. To the most extreme end, even for another type of life (different from the “life as we know it” on the Earth)—if possibly existing, these analyses would also be useful as long as our concept of life defined in terms of “Darwinian evolution” and “self-sustainment” stands.

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
