# Peer review of "What Does “the RNA World” Mean to “the Origin of Life”?"

_life, 2017, doi:10.3390/life7040049_

Round 1

Reviewer 1 Report

The analysis presented in this paper is most welcome because it looks at the logic behind the RNA World hypothesis rather than arguing for its plausibility without questioning the adequacy of its basic assumptions as an explanation for the origin of life.

This paper is suitable for publication in Life subject to some clarification and attention being paid to some significant omissions.

My first reaction when I read the abstract and agreed to act as a referee was that some points about self-sustaining were at variance with some of my own recent findings, but when I received the whole text I modified my point of view.  While I thought I would not have much to say, I found the paper very engaging and have attempted to be as helpful as I can be to the author in making minor improvements.

At the very beginning of the abstract I think it would be more appropriate to describe Darwinian evolution and self-sustaining as "two of life's distinct aspects", rather than implying that these two aspects offer a sufficient defintion of life, even at its earliest stages.  Although the paper has a brief discussion of what "self-" means, no distinction is made between descriptions couched in terms of physics and chemistry (events only HAPPEN in this world) and those couched in terms of biology (things are DONE in this world; there is agency at a level higher than the laws of nature).

I think it has to be acknowledged that the priority given to genetic information over self-sustaining (as if "you can't have the latter without the former") is a plausible but unproven assumption.  The possibility of the simultaneous co-dependent emergence of the two together is also a possibility (as outlined in my own work: see below).  In the sentence of lines 67-69 the author comes very close to the idea that life requires inheritance of a (self-sustaining) "interpreter" of the genetic information.  The sentence on lines 75-78 begs the question of what degree of autonomy, what degree of organisation the "self-sustaining" needs.  Could you just transfer the genes back into the system and expect it to still function as normal?  Or do the non-genetic components have to be organised, not just in terms of relative concentrations and confinement, but also in terms of internal spatial arrangement of the molecules?  Would such order have to be created among the components before you could put the genes back in and expect the system to function and survive?  So, can you attribute the causation to the genes as "instructors"?  That is why I say that the sentence on lines 90-93 should be portrayed as an premise, not something about which we have 100% certainty.  (I personally find the simultaneous emergence version closer more likely to produce what is observed in molecular biological systems, not the genes first version.)

The sentence lines 127-129 is a breath of fresh air in discussions of the RNA World.

Lines 134-136 introduce an idea of "encoding" without discussing the fact that this has often been posed as THE central question for the origin of life: the catalytic properties of RNA were grabbed as a facile solution to this problem concerning the emergence of an informatically regular mapping from genotype to phenotype in biology.  The mapping from RNA sequences onto phenotypic properties is nothing like the mapping from nucleic acids onto proteins. The essence of molecular biology is a informatic mapping, not a physico-chemical mapping, providing molecular biological systems with a degree of computational control that transcends physico-chemical determination.  The molecular biological mapping needs the components it makes to produce the mapping.  In my view this is the main "self-sustaining" that distinguishes biology from other non-thermodynamic systems.  The RNA world completely lacks this sort of self-sustaining because the genotype to phenotype mapping requires only the chemistry of RNA folding, not special system components.

Any thermodynamic system that is in a state far from equilibrium is a self-sustaining system. Prigogine started writing about this, and its relationship to biology, 50 years ago.  So, I would be much happier if the two (very nice) sentences in lines 155-158 included a statement to the effect that the simultaneity of genetic information and function could be the source of a self-sustaining that was much more than thermodynamic.  I have written about this possibility quite extensively and my two recent papers with Charlie Carter relate this to the RNA world hypothesis (see below).  It would be appropriate to add some ackowledgment of this work in lines 180 and 194.

It is very refreshing to read a paper about the RNA world which refers to the need for some kind of spatial organisation (like reaction-diffusuin coupling or compartmentation) or "tagging" to overcome the problem of parasitism.  The main discussion in Section 3 of the paper justifies its publication in Life, but what is evident at the end of Section 3.3 (lines 382-384) is that previous discussion of the RNA World has been focussed almost entirely on bringing together various material components, not how the information contained in the genetic polymer may actually be USED AS INFORMATION by the system to sustain itself.

The sentence in lines 393-395 needs to be justified in relation to Eigen's threhold criterion.  One can only say that "it could have spread in the system" if it operated ("sustained itself"!) above the error threshold for its own replication.

Then, at lines 435-436, the assertion "the chromosome, as an RNA molecule itself, serves as the ultimate target of natural selection" can only refer to true Darwinian selection if the genotype to phenotype mapping is somehow fixed.  Already at this stage of evolution, surely the ultimate target of selection is the co-dependent unit of chromosome and functional components and the latter may have already attained a meta-stable, far-from-equilibrium dynamic state which was semi-autonomous in response to environmental boundary conditions (selection pressure).  A comment on this point would be welcome here.

I would dispute the generality of the claim (lines 445-446) that it was through the invention of linked genes on chromosomes "that our living world achieved its fundamental potential to become complicated".  I do not dispute the importance of chromosomally linking genes, just the relative importance of this compared to other factors.  It would be good to see other views acknowledged, like those expressed by Charlie Carter and me, that the acquisition of computational capability (information-processing beyond copying) was much more fundamental to the evolution of functional complexity, as seen in the evolution of coding.

Once again, the first paragraph of section 3.5 (lines 448-455) would benefit from consideration of the work of Eigen.  For any "master molecule" to survive, it must be "kinetically self-sustaining" in the sense that it must have the property of being able to keep itself, in the presence of its quasi-species mutant distribution, above the error threshold.  And then again, in the next sentence (lines 456-460), RAF systems are self-sustaining in the sense of making all of their necessary components from the food set, and they must likewise be in some dynamic configuration that holds them away from the basin of attraction of extinction.  The next paragraph (lines 461-469) add a new dimension, but relating it back to the discussion of the meaning of "self-" in lines 55-88 would be helpful.  Considering lines 461-500, is it being asserted that "self-" can only refer to a system that is spatially enclosed in a membrane, reminiscent of Gánti's chemoton?

Line 524 talks about "a different one [scenario]".  Reference to the alternative quite detailed elaboration in the two Carter/Wills papers would be appropriate here.  The "succinct" world discussed in this paper misses what these other authors have taken to be what is absolutely essential to any consideration of the origin of "life", that is, a kind of informatic (or "computational") self-sustaining which involves a reflexivity in information processing just as profound as the reflexivity in reaction chemistry required for thermodynamic self-sustaining.  However, I wish to compliment the author for ending with such a modest and sensible conclusion.

The Carter/Wills papers I have referred to are both now saccepted for publication:

Carter C W Jr, Wills P R. Interdependence, Reflexivity, Fidelity, Impedance Matching, and the Evolution of Genetic Coding. Mol. Biol. Evol. (2017) https://doi.org/10.1101/139139

Wills P R, Carter C W Jr. Insuperable problems of the genetic code initially emerging in an RNA World. BioSystems (2017) DOI: 10.1016/j.biosystems.2017.09.006

I also offer the following indicators of problems in the use of English.  The list is probably incomplete.

39 talked

89 concerned about

208 obtain

274 no matter WHAT the membrane is composed of, [COMMA]

275 sufficiently

434 cellular

444 introducing

452 stays unchanged

521 scenario has

Overall I think this is an excellent, thoughtful, helpful paper about the logic of the RNA world.

Author Response

I am grateful to the reviewer for his thoughtful comments on this paper. The following is my response to these comments.

The analysis presented in this paper is most welcome because it looks at the logic behind the RNA World hypothesis rather than arguing for its plausibility without questioning the adequacy of its basic assumptions as an explanation for the origin of life.

This paper is suitable for publication in Life subject to some clarification and attention being paid to some significant omissions.

My first reaction when I read the abstract and agreed to act as a referee was that some points about self-sustaining were at variance with some of my own recent findings, but when I received the whole text I modified my point of view.  While I thought I would not have much to say, I found the paper very engaging and have attempted to be as helpful as I can be to the author in making minor improvements.

--- I appreciate the reviewer’s endorsement.

At the very beginning of the abstract I think it would be more appropriate to describe Darwinian evolution and self-sustaining as "two of life's distinct aspects", rather than implying that these two aspects offer a sufficient definition of life, even at its earliest stages. Although the paper has a brief discussion of what "self-" means, no distinction is made between descriptions couched in terms of physics and chemistry (events only HAPPEN in this world) and those couched in terms of biology (things are DONE in this world; there is agency at a level higher than the laws of nature).

--- I am not quite certain about what the reviewer means. I do think that they are the two fundamental aspects of life. Indeed, due to the confusions which are brought about when people talk about the two aspects together, we had better splitting the definition of life (see a previous paper of mine for a discussion in detail [3]). However, I do not think the intention to incorporate other aspects into the definition, which would make the definition further complicated, is appropriate.

I think it has to be acknowledged that the priority given to genetic information over self-sustaining (as if "you can't have the latter without the former") is a plausible but unproven assumption. The possibility of the simultaneous co-dependent emergence of the two together is also a possibility (as outlined in my own work: see below). 

--- Yes, I think so. What I mean is that self-sustaining cannot arise before the advent of genetic information. As explained in the text of the paper, even if some kind of self-sustaining appeared before the advent of genetic information, it was not that feature of self-sustaining as we mean relevant to the life phenomenon.

In the sentence of lines 67-69 the author comes very close to the idea that life requires inheritance of a (self-sustaining) "interpreter" of the genetic information. The sentence on lines 75-78 begs the question of what degree of autonomy, what degree of organisation the "self-sustaining" needs.  Could you just transfer the genes back into the system and expect it to still function as normal?  Or do the non-genetic components have to be organised, not just in terms of relative concentrations and confinement, but also in terms of internal spatial arrangement of the molecules?  Would such order have to be created among the components before you could put the genes back in and expect the system to function and survive?  So, can you attribute the causation to the genes as "instructors"?  That is why I say that the sentence on lines 90-93 should be portrayed as an premise, not something about which we have 100% certainty.  (I personally find the simultaneous emergence version closer more likely to produce what is observed in molecular biological systems, not the genes first version.)

--- Yes, I have added a sentence in the place, indicating that we cannot rule out the possibility for the simultaneous advent of the two aspects. In the scenario concerning the RNA world, for instance, if the RNA world began with an RNA-based protocell, instead of with some ribozyme in a ‘naked’ scene, the two features would emerge together. Whether an RNA/peptide world is convincing is another issue that will be discussed later. But here I would like to note that, according to the series of our computer simulation studies on the origin and early development of the RNA world [28, 52, 68,84,89,92 and some others not listed here], we have confirmed that, at least in terms of evolutionary dynamics, RNA alone could work well as both genes and functional molecules, supporting the “primitive living world”. For instance, an RNA-based protocell may not need a very high degree of autonomy or organisation to be self-sustaining and to engage in Darwinian evolution.

The sentence lines 127-129 is a breath of fresh air in discussions of the RNA World.

Lines 134-136 introduce an idea of "encoding" without discussing the fact that this has often been posed as THE central question for the origin of life: the catalytic properties of RNA were grabbed as a facile solution to this problem concerning the emergence of an informatically regular mapping from genotype to phenotype in biology. The mapping from RNA sequences onto phenotypic properties is nothing like the mapping from nucleic acids onto proteins. The essence of molecular biology is an informatic mapping, not a physico-chemical mapping, providing molecular biological systems with a degree of computational control that transcends physico-chemical determination.  The molecular biological mapping needs the components it makes to produce the mapping.  In my view this is the main "self-sustaining" that distinguishes biology from other non-thermodynamic systems. The RNA world completely lacks this sort of self-sustaining because the genotype to phenotype mapping requires only the chemistry of RNA folding, not special system components.

--- To this point, I would like to say that, in modern living world, the mapping should include both the informatic mapping from nucleic acids onto proteins and the physico-chemical mapping regarding the folding of the proteins; in the RNA world, the mapping should also have included both the informatic mapping – from RNA genes to ribozymes (simply via template-directed copying, the same way as it replicated) and the physico-chemical mapping regarding the folding of the ribozymes. No matter how, here I introduce the notion of “encoding” just to represent the informatic mapping (“instructing the synthesis of”) mentioned by the reviewer. I have added some annotations in the corresponding place. Thanks for the comment.

Any thermodynamic system that is in a state far from equilibrium is a self-sustaining system. Prigogine started writing about this, and its relationship to biology, 50 years ago. So, I would be much happier if the two (very nice) sentences in lines 155-158 included a statement to the effect that the simultaneity of genetic information and function could be the source of a self-sustaining that was much more than thermodynamic. I have written about this possibility quite extensively and my two recent papers with Charlie Carter relate this to the RNA world hypothesis (see below).  It would be appropriate to add some acknowledgment of this work in lines 180 and 194.

--- I generally agree with the reviewer in this point. Indeed, the simultaneity of genetic information and function could be the source of a self-sustaining that was much more than an ordinary Prigogine system can provide. That is why an ordinary Prigogine system cannot engage in Darwinian evolution whereas a living entity can. But I think it is inappropriate to exclude living entities from thermodynamic systems – living entities should be only peculiar far-from-equilibrium thermodynamic systems which Prigogine meant. In addition, the two papers mentioned here by the reviewer have been cited in the text, where the RNA/peptides world is discussed [45,46].

It is very refreshing to read a paper about the RNA world which refers to the need for some kind of spatial organisation (like reaction-diffusion coupling or compartmentation) or "tagging" to overcome the problem of parasitism. The main discussion in Section 3 of the paper justifies its publication in Life, but what is evident at the end of Section 3.3 (lines 382-384) is that previous discussion of the RNA World has been focused almost entirely on bringing together various material components, not how the information contained in the genetic polymer may actually be USED AS INFORMATION by the system to sustain itself.

--- Yes, I generally agree.

The sentence in lines 393-395 needs to be justified in relation to Eigen's threshold criterion. One can only say that "it could have spread in the system" if it operated ("sustained itself"!) above the error threshold for its own replication.

--- Yes, I have added the annotation concerning the error threshold. Thanks for the comment. However, I should note here that this is not a context for the expression “sustained itself”, which may be confused with the self-sustaining we discussed here. The phrase “sustained itself” here is concerned to the existence of the molecular form in the system but not for the existence of the molecule itself as an entity. But the self-sustaining we focus on here is for the existence of an entity. Particular attention should be paid to the difference between a form and an entity when we intend to understand life phenomena (see my concept paper “the Essence of Life” [3] for a detailed discussion).

Then, at lines 435-436, the assertion "the chromosome, as an RNA molecule itself, serves as the ultimate target of natural selection" can only refer to true Darwinian selection if the genotype to phenotype mapping is somehow fixed. Already at this stage of evolution, surely the ultimate target of selection is the co-dependent unit of chromosome and functional components and the latter may have already attained a meta-stable, far-from-equilibrium dynamic state which was semi-autonomous in response to environmental boundary conditions (selection pressure). A comment on this point would be welcome here.

--- I agree with the reviewer. However, here, what I mean is that the “ultimate” target of selection is concerning the genome, whereas the “direct” target of selection is concerning the phenotype.

I would dispute the generality of the claim (lines 445-446) that it was through the invention of linked genes on chromosomes "that our living world achieved its fundamental potential to become complicated".  I do not dispute the importance of chromosomally linking genes, just the relative importance of this compared to other factors. It would be good to see other views acknowledged, like those expressed by Charlie Carter and me, that the acquisition of computational capability (information-processing beyond copying) was much more fundamental to the evolution of functional complexity, as seen in the evolution of coding.

--- Indeed, it should be acknowledged that the emergence of translation opened the vast possibilities to explore the sequence-function space of proteins in natural selection, which is crucial to the prosperity of our living world. This is obvious. But chromosomally linking genes, as a fundamental event whose significance appears not so obvious to us, also deserve highlighting. The two assertions do not contradict with each other. In addition, I think that the acquisition of “computational capability” is important only on account of the importance of proteins. Just imagine that if RNA can in itself be so “omnipotent” as proteins to act as functional molecules — then no information-process beyond copying are needed, and our extant living world might still exist as an RNA world or at most, a DNA/RNA world. Thus, it seems that the acquisition of computational capability was not so “fundamental” as the reviewer thinks. Because both the reviewer’s comments and my response would be visible to readers, I leave these ideas here for their reference.

Once again, the first paragraph of section 3.5 (lines 448-455) would benefit from consideration of the work of Eigen. For any "master molecule" to survive, it must be "kinetically self-sustaining" in the sense that it must have the property of being able to keep itself, in the presence of its quasi-species mutant distribution, above the error threshold.  And then again, in the next sentence (lines 456-460), RAF systems are self-sustaining in the sense of making all of their necessary components from the food set, and they must likewise be in some dynamic configuration that holds them away from the basin of attraction of extinction.  The next paragraph (lines 461-469) add a new dimension, but relating it back to the discussion of the meaning of "self-" in lines 55-88 would be helpful. Considering lines 461-500, is it being asserted that "self-" can only refer to a system that is spatially enclosed in a membrane, reminiscent of Gánti's chemoton?

--- This is a misunderstanding. As mentioned above, the self-sustaining we focus here is the existence of an entity instead of the existence of a form (i.e., a kind of entity). This is a sort of “horizontal self-sustaining” instead of “vertical self-sustaining”. The “vertical self-sustaining” is concerned with the ongoing of Darwinian evolution. The meaning of the “self-sustaining” involved in the definition of life should be only the “horizontal self-sustaining”. Therefore, in the “naked scene”, “self-“ should refer to a molecule; when Darwinian entities became protocells, “self-“ should refer to a system spatially enclosed in the membrane (indeed, like Gánti's chemoton). Indeed, the origin of life is a field difficult to study and understand, and the communication in the field is usually full of misunderstandings. Thank to the policy of this journal which allows readers to access the reviewer’s comments and author’s response together with the manuscript, we have the chance to illustrate these key concepts in a more clear way.

Line 524 talks about "a different one [scenario]".  Reference to the alternative quite detailed elaboration in the two Carter/Wills papers would be appropriate here. The "succinct" world discussed in this paper misses what these other authors have taken to be what is absolutely essential to any consideration of the origin of "life", that is, a kind of informatic (or "computational") self-sustaining which involves a reflexivity in information processing just as profound as the reflexivity in reaction chemistry required for thermodynamic self-sustaining.  However, I wish to compliment the author for ending with such a modest and sensible conclusion.

--- I have cited the two Carter/Wills paper above [45,46]. To keep this conclusion remark concise, I think it better not to cite any references here. As an additional remark, from this comment and those above I notice that the reviewer seems seriously doubt about the plausibility of the origin of life without a kind of informatic mapping like genetic coding (nucleic acids to proteins). But as mentioned above, according to our relevant studies by computer simulation [28, 52, 68,84,89,92 and some others not listed here], the RNA world, which lacked such a kind of “computational process“, may have booted up well and even developed to a rather complex level, e.g., that of protocells with chromosomally linked genes. That is, either Darwinian evolution or self-sustaining (of the protocells) therein is “well-demonstrated”. I confirm here that these computer simulation studies are quite robust and can demonstrate the plausibility of the RNA world at least in terms of evolutionary dynamics.

The Carter/Wills papers I have referred to are both now accepted for publication:

Carter C W Jr, Wills P R. Interdependence, Reflexivity, Fidelity, Impedance Matching, and the Evolution of Genetic Coding. Mol. Biol. Evol. (2017) https://doi.org/10.1101/139139

Wills P R, Carter C W Jr. Insuperable problems of the genetic code initially emerging in an RNA World. BioSystems (2017) DOI: 10.1016/j.biosystems.2017.09.006

--- Yes, I have cited the two papers [45,46].

I also offer the following indicators of problems in the use of English. The list is probably incomplete.

39 talked

89 concerned about

208 obtain

274 no matter WHAT the membrane is composed of, [COMMA]

275 sufficiently

434 cellular

444 introducing

452 stays unchanged

521 scenario has

--- I have changed the words mentioned here and have made a thorough check upon the whole manuscript. Many thanks.

Overall I think this is an excellent, thoughtful, helpful paper about the logic of the RNA world.

--- Once again, I thank to the reviewer for his detailed, helpful comments.

Reviewer 2 Report

The manuscript entitled “What does ‘the RNA world’ mean to ‘the origin of life’” discusses the two main aspects of the definition of life: Darwinian evolution and self-sustainability. Importantly, the author describes the importance in separating these two concepts, and also the fact that the only relevant self-sustaining biological systems are those which have already developed the capability of Darwinian evolution. This, of course, is in the context of an RNA system, which is believed to be the first biomolecule of life. Specifically, if a self-sustaining system were (i.e., an autocatalytic system) to have developed before the emergence of Darwinian evolution-controlled RNA system, then how would this autocatalytic system have been linked to the genetic control of the RNA system? Would the autocatalytic system even have been able to couple to the RNA system? This is a major question, and rather it may have been more likely for any living system (i.e., RNA) to have developed self-sustainability only after the emergence of the capability of Darwinian evolution. Specifically, the origin of life should mean the origin of Darwinian evolution, which is labeled by the emergence of genetic material. This is a very important point that the manuscript describes well, and other researchers in the origins field should also keep in mind when studying self-sustainable or autocatalytic systems. The author then goes through a detailed description of the plausibility of the RNA world scenario of the origin of life, including the assembly of the first RNA monomers, then subsequently polymers, the emergence of the first RNA replicase, problems with parasites, the emergence of the first protocells, the emergence of Darwinian evolution only after the emergence of protocells and replicases, cooperativity of ribozymes, all culminating with the development of self-sustainability.

Although the ideas and arguments in the manuscript are generally well-presented, and the main topic discussed is quite important for those in the origins field, I do have a few comments to discuss.

Major comment 1) The English grammar in this manuscript is generally not good. There are many cases of misused and incorrectly-used words, especially different verb tenses and noun forms. Specifically, the term “self-sustaining”, which is used quite often in this manuscript and is a central part of the scientific idea, is incorrectly used. Rather, the author should use “self-sustainability” or something similar as “self-sustaining” is an adjective and not a noun. There are many other grammar errors that would be too long to list here. Additionally, there are many informal colloquialisms used, such as “got” (Line 18 195, etc.) that should not be used in a scientific publication. Please fix all language-based errors.

Minor Comment 1) Generally, it is believed by many in the origins field that the only relevant definition of “life” is extant life as we know it today, which contains proteins, lipids, nucleic acids, etc. Although it is reasonable to perform origin of life research based on our understanding of current life, one must still consider that our notion of life is only one of many possibilities of life. For example, are nucleic acids the only possible genetic systems? This question also confounds those working in the life detection technology fields, where although we only know of one type of “life” and much of life detection technology is based on this known type of life, there is still uncertainty as to what type of life would have emerged in other non-Earth environments (with different atmospheres, chemical compositions, temperatures, etc.). Thus, perhaps it would be reasonable to include a statement somewhere that states that although much of origins research has focused on systems which resemble extant life, it is not the only possibility for life. That perhaps if life had evolved in a different way, then it may not have been necessary for the origin of self-sustainability to strictly proceed that of the origin of a genetic system capable of Darwinian evolution. This is also part of the reason that drives certain researchers to focus on self-sustaining replication systems even in the absence of a genetic system capable of Darwinian evolution.

Minor Comment 2) The author comments around lines 177-195 that the RNA/peptide world scenario is flawed due partially to the fact that peptides in the environment would have been random and thus would not have assisted in evolution. Although this may be a valid claim, there may have been certain scenarios by which certain peptide sequences could have preferentially aggregated to high amounts, including through sequence-preferential binding to mineral surfaces (Segvich, et al, 2009: https://www.ncbi.nlm.nih.gov/pmc/articles/PMC2744811/), sequence-preferential formation of coacervates (Aumiller & Keating, 2016: https://www.nature.com/nchem/journal/v8/n2/full/nchem.2414.html) or other two-phase systems (Barros et al, 2014: http://www.sciencedirect.com/science/article/pii/S0021967313018797), or reactions in non-aqueous solvents (Hayashi et al, 1965: http://www.tandfonline.com/doi/abs/10.1080/00021369.1966.10858601), among others. The author may briefly mention each of these systems as possible (but still each obviously somewhat flawed) methods to enrich specific peptide sequences.

Additionally, although it has been demonstrated that RNA polymerase ribozymes can exist, there has still not been clear evidence showing the plausibility of the assembly of such an RNA polymerase ribozyme without the aid of itself, similar to a bootstrap paradox. Mostly, this research has been performed on RNA-only systems. Thus researchers have started focusing on non-purely RNA systems partially due to this fact.

Minor Comment 3) The author in section 3.1 comments on permeability of nucleotides and other nutrients being a major problem for lipid-based protocell systems. Certain coacervate systems do not have this problem, and in fact significantly segregate and concentrate important nutrients including nucleotides and cationic metal ions (Frankel et al, 2016: http://pubs.acs.org/doi/abs/10.1021/acs.langmuir.5b04462) or even minerals (Pir Cakmak & Keating, 2017: https://www.nature.com/articles/s41598-017-03033-z). However, there are still some issues with stable compartmentalization of genetic polymers within certain coacervate systems (Jia et al, 2014: https://www.ncbi.nlm.nih.gov/pubmed/24577897). Perhaps the author may make a comment on these alternative encapsulation systems that, although again somewhat flawed due to certain systems’ potential fast exchange with external environments, may have at least provided a possible solution to the nutrient permeability problem.

Minor Comment 4) About parasites, in addition to the theoretical experiments within protocells (Section 3.3), there have also been lab experiments as well (Bansho et al, 2016: http://www.pnas.org/content/113/15/4045.short).

Minor Comment 5) In Line 438, the author mentions the “one molecule, one function” rule. Although this is generally accepted, especially in prebiotic systems, there are still certain prebiotically-relevant molecules, such as ribozymes, that have multiple simultaneous functions (Landweber & Pokrovskaya, 1999: http://www.pnas.org/content/96/1/173.full.pdf). In fact, perhaps it could be considered that all sequence-specific ribozymes have dual functions, as sequence-recognition could be a completely separate process from the catalytic activity that it affords.

Author Response

I am grateful to the reviewer for his thoughtful comments on this paper. The following is my response to these comments.

The manuscript entitled “What does ‘the RNA world’ mean to ‘the origin of life’” discusses the two main aspects of the definition of life: Darwinian evolution and self-sustainability. Importantly, the author describes the importance in separating these two concepts, and also the fact that the only relevant self-sustaining biological systems are those which have already developed the capability of Darwinian evolution.

--- Yes, the reviewer’s interpretation here is quite right.

This, of course, is in the context of an RNA system, which is believed to be the first biomolecule of life. Specifically, if a self-sustaining system were (i.e., an autocatalytic system) to have developed before the emergence of Darwinian evolution-controlled RNA system, then how would this autocatalytic system have been linked to the genetic control of the RNA system? Would the autocatalytic system even have been able to couple to the RNA system? This is a major question, and rather it may have been more likely for any living system (i.e., RNA) to have developed self-sustainability only after the emergence of the capability of Darwinian evolution.

--- Yes, the answers of these questions remain unclear. I do not intend to discuss them in this paper. Personally, I do think it quite unlikely that a precursive autocatalytic system could be linked to or couple to the genetic control of an RNA system, which emerged later. To avoid the need to discuss these questions in depth, I only indicated in the paper that even if some kind of self-sustaining appeared before the advent of genetic information, it was not that feature of self-sustaining as we mean relevant to the life phenomenon

Specifically, the origin of life should mean the origin of Darwinian evolution, which is labeled by the emergence of genetic material. This is a very important point that the manuscript describes well, and other researchers in the origins field should also keep in mind when studying self-sustainable or autocatalytic systems. The author then goes through a detailed description of the plausibility of the RNA world scenario of the origin of life, including the assembly of the first RNA monomers, then subsequently polymers, the emergence of the first RNA replicase, problems with parasites, the emergence of the first protocells, the emergence of Darwinian evolution only after the emergence of protocells and replicases, cooperativity of ribozymes, all culminating with the development of self-sustainability.

 ---Yes, the reviewer’s understanding on this paper is unexpectedly “elegant”.

Although the ideas and arguments in the manuscript are generally well-presented, and the main topic discussed is quite important for those in the origins field, I do have a few comments to discuss.

         Major comment 1) The English grammar in this manuscript is generally not good. There are many cases of misused and incorrectly-used words, especially different verb tenses and noun forms. Specifically, the term “self-sustaining”, which is used quite often in this manuscript and is a central part of the scientific idea, is incorrectly used. Rather, the author should use “self-sustainability” or something similar as “self-sustaining” is an adjective and not a noun. There are many other grammar errors that would be too long to list here. Additionally, there are many informal colloquialisms used, such as “got” (Line 18 195, etc.) that should not be used in a scientific publication. Please fix all language-based errors.

--- Here, “self-sustaining” is used as a gerund, which I thought should be equivalent to a noun. I am not quite clear what the reviewer meant here. If I have to choose a noun form, I think it should be “self-sustainment”. If we say one aspect of life is self-sustainability, correspondingly, we have to depict the other aspect of life as the ability of undergoing Darwinian evolution, which seems too verbose. I have fixed some language-based errors according to the opinions of another reviewer and have check the whole manuscript carefully to improve the language. If the reviewer can point out other errors, I will be grateful to him for his kindness.

Minor Comment 1) Generally, it is believed by many in the origins field that the only relevant definition of “life” is extant life as we know it today, which contains proteins, lipids, nucleic acids, etc. Although it is reasonable to perform origin of life research based on our understanding of current life, one must still consider that our notion of life is only one of many possibilities of life. For example, are nucleic acids the only possible genetic systems? This question also confounds those working in the life detection technology fields, where although we only know of one type of “life” and much of life detection technology is based on this known type of life, there is still uncertainty as to what type of life would have emerged in other non-Earth environments (with different atmospheres, chemical compositions, temperatures, etc.). Thus, perhaps it would be reasonable to include a statement somewhere that states that although much of origins research has focused on systems which resemble extant life, it is not the only possibility for life. That perhaps if life had evolved in a different way, then it may not have been necessary for the origin of self-sustainability to strictly proceed that of the origin of a genetic system capable of Darwinian evolution. This is also part of the reason that drives certain researchers to focus on self-sustaining replication systems even in the absence of a genetic system capable of Darwinian evolution.

--- Yes, I have add a relevant sentence at the end of the paper. Thanks. But let me talk more at this point. Indeed, in principle, life can be of another type. That is why we should not define life only in terms of materials it is based on, but using words like “Darwinian evolution” and “self-sustaining”. Any forms which may fall into this frame should be called “life”. The way to implement these two aspects, however, are quite particular, which points to three fundamental mechanisms: “modular replication” of a heterogenous polymer, sequence-dependent folding of this polymer or another type of material encoded by this polymer to engendering special functions, and the assembly of membrane to enclose these polymers (see the second part of my paper “The essence of life” [3] for details). All these requirements seems to be very unlikely to be met beyond the scenario we see in our extant living world on the Earth. Therefore, we ought to acknowledge that to perform origin of life research based on our understanding of current life should be, at least largely, on the right way. Finally, it should be stressed that just according the logic presented in the analysis of this manuscript, “self-sustaining replication systems in the absence of a genetic system” should have little relevance to life phenomena, not matter what type of life it is. That is just a case reflecting the spirit of this manuscript, as well as that of the previous paper regarding the essence of life [3] – recurring to the “life as we know it” in discussion but bearing a significance concerning the general concept of “life”.

Minor Comment 2) The author comments around lines 177-195 that the RNA/peptide world scenario is flawed due partially to the fact that peptides in the environment would have been random and thus would not have assisted in evolution. Although this may be a valid claim, there may have been certain scenarios by which certain peptide sequences could have preferentially aggregated to high amounts, including through sequence-preferential binding to mineral surfaces [37], sequence-preferential formation of coacervates [39] or other two-phase systems [38], or reactions in non-aqueous solvents [36], among others. The author may briefly mention each of these systems as possible (but still each obviously somewhat flawed) methods to enrich specific peptide sequences.

--- Yes, I have mentioned them. Note that the key issue is not the specificity of the peptide sequences but the “informatic mapping” (as mentioned in another reviewer’s report) between RNA and the peptides.

Additionally, although it has been demonstrated that RNA polymerase ribozymes can exist, there has still not been clear evidence showing the plausibility of the assembly of such an RNA polymerase ribozyme without the aid of itself, similar to a bootstrap paradox. Mostly, this research has been performed on RNA-only systems. Thus researchers have started focusing on non-purely RNA systems partially due to this fact.

--- Yes, that is the fact. However, as mentioned in the manuscript, “Now that the tail-binding of the ribozyme onto the template can greatly improve the ribozyme’s efficiency, it would be naturally to expect that a shorter version of the replicase (perhaps about 30-50 nt long), with the tail-binding retained, may have had an efficiency, albeit somewhat lower, yet enough to support its own thriving in the system”. A 30-50 nt long RNA may have assembled, for instance, under mineral catalysis [58, 94].

Minor Comment 3) The author in section 3.1 comments on permeability of nucleotides and other nutrients being a major problem for lipid-based protocell systems. Certain coacervate systems do not have this problem, and in fact significantly segregate and concentrate important nutrients including nucleotides and cationic metal ions (Frankel et al, 2016: http://pubs.acs.org/doi/abs/10.1021/acs.langmuir.5b04462) or even minerals (Pir Cakmak & Keating, 2017: https://www.nature.com/articles/s41598-017-03033-z). However, there are still some issues with stable compartmentalization of genetic polymers within certain coacervate systems (Jia et al, 2014: https://www.ncbi.nlm.nih.gov/pubmed/24577897). Perhaps the author may make a comment on these alternative encapsulation systems that, although again somewhat flawed due to certain systems’ potential fast exchange with external environments, may have at least provided a possible solution to the nutrient permeability problem.

--- Yes, I generally agree with the reviewer in this point. However, if these alterative encapsulation systems are introduced here, a simple comment on them seems to be not sufficient. To be more focus in this paper, I would like to leave these opinions here, considering readers can access the text of the reviewers’ comments and the authors’ response according to the journal’s policy.

Minor Comment 4) About parasites, in addition to the theoretical experiments within protocells (Section 3.3), there have also been lab experiments as well [70].

---Yes, I have mentioned this work. Thanks.

Minor Comment 5) In Line 438, the author mentions the “one molecule, one function” rule. Although this is generally accepted, especially in prebiotic systems, there are still certain prebiotically-relevant molecules, such as ribozymes, that have multiple simultaneous functions (Landweber & Pokrovskaya, 1999: http://www.pnas.org/content/96/1/173.full.pdf). In fact, perhaps it could be considered that all sequence-specific ribozymes have dual functions, as sequence-recognition could be a completely separate process from the catalytic activity that it affords.

---Yes, I agree with the reviewer concerning the dual functions for a ribozyme. However, here what I meant is that one functional molecule, generally, can engage into one special active event. For example, we cannot expect that an RNA replicase ribozyme can, in addition to catalyzing the template-directed copying of RNA, also catalyze the synthesis of nucleotides and the synthesis of lipid molecules, which should be accomplished by two other ribozymes, i.e., the nucleotide synthetase ribozyme and the amphiphilic molecule synthetase ribozyme, as mentioned in the manuscript. Even though it is yet possible that a protein molecule or an RNA molecule could engage into two special active events, the key point here is that we cannot expect that one functional molecule can carry more and more functions in the evolution towards complexity. I have added an annotation in the text. The sentence now reads “due to the golden rule ‘one molecule, one function’ (e.g., one enzyme or one ribozyme, typically, could only catalyze one reaction), a single molecule seems far from being sufficient to bear multiple phenotypes of a complex life form”. Many thanks.

Reviewer 3 Report

   The manuscript is aimed at providing a reassessment of the relevance of the RNA world hypothesis to the origin of life. Its main tenets lie in the need for a replication process involving base pairing to allow variation and therefore Darwinian evolution through the selection of the best replicating entities. It constitutes a piece of work gathering information from the literature in the perspective of a top-down approach to the origin of life (eventually considering the part played by RNA in biology to deduce some conclusions on the origin of life process). It starts from a life definition separating the concepts of Darwinian evolution and of self-sustaining abilities. Though I consider that most of the data selected by the author to support his views are consistent with an important role of RNA at the early stages of biology and possibly in the origin of life, I have some concerns about this manuscript that in my opinion just constitutes a reformulation of the “molecular biologist’s dream” as quoted by Jerry Joyce and Leslie Orgel, who however associated this statement with that of the “prebiotic chemist’s nightmare”. Though they clearly show that RNA should not be dismissed from the origin of life process, I do not consider that the progresses made in the past decade in the synthesis of nucleotides support that the availability of activated nucleotide monomers can be taken for granted. The presence of an environment harbouring a concentrated mixture of activated RNA monomers in which replicase ribozymes may emerge and evolve remains exceedingly unlikely. It is meaningful that the recent experiments of John Sutherland’s group on the synthesis of nucleotides points towards a process that produces at the same time amino acids and phospholipid precursors suggesting that it is not possible to separate one class of biomolecules in the origin of life process. In addition, to the best of my knowledge, a prebiotically reliable chemical process able to activate monomers and to provide the driving force for polymerization remains to be discovered. We can therefore conclude that the process should have been chemically more complex (involving cooperation between classes of biomolecules and biopolymers) than what can be deduced from a conceptual approach of biology and I urge the author to take into account the role of the chemical behaviour of the components. Recent developments in systems chemistry have been made giving an idea of how the metabolic aspect of the problem (related to self-sustaining aspects) could be intimately associated with the emergence of replicators (not only by encapsulating ribozymes in compartments).

Author Response

I am grateful to the reviewer for his thoughtful comments on this paper. The following is my response to these comments.

The manuscript is aimed at providing a reassessment of the relevance of the RNA world hypothesis to the origin of life. Its main tenets lie in the need for a replication process involving base pairing to allow variation and therefore Darwinian evolution through the selection of the best replicating entities. It constitutes a piece of work gathering information from the literature in the perspective of a top-down approach to the origin of life (eventually considering the part played by RNA in biology to deduce some conclusions on the origin of life process). It starts from a life definition separating the concepts of Darwinian evolution and of self-sustaining abilities.

--- Yes, thanks.

Though I consider that most of the data selected by the author to support his views are consistent with an important role of RNA at the early stages of biology and possibly in the origin of life, I have some concerns about this manuscript that in my opinion just constitutes a reformulation of the “molecular biologist’s dream” as quoted by Jerry Joyce and Leslie Orgel, who however associated this statement with that of the “prebiotic chemist’s nightmare”. Though they clearly show that RNA should not be dismissed from the origin of life process, I do not consider that the progresses made in the past decade in the synthesis of nucleotides support that the availability of activated nucleotide monomers can be taken for granted. The presence of an environment harbouring a concentrated mixture of activated RNA monomers in which replicase ribozymes may emerge and evolve remains exceedingly unlikely. It is meaningful that the recent experiments of John Sutherland’s group on the synthesis of nucleotides points towards a process that produces at the same time amino acids and phospholipid precursors suggesting that it is not possible to separate one class of biomolecules in the origin of life process. In addition, to the best of my knowledge, a prebiotically reliable chemical process able to activate monomers and to provide the driving force for polymerization remains to be discovered. We can therefore conclude that the process should have been chemically more complex (involving cooperation between classes of biomolecules and biopolymers) than what can be deduced from a conceptual approach of biology and I urge the author to take into account the role of the chemical behaviour of the components. Recent developments in systems chemistry have been made giving an idea of how the metabolic aspect of the problem (related to self-sustaining aspects) could be intimately associated with the emergence of replicators (not only by encapsulating ribozymes in compartments).

--- Yes, the prebiotic pool may have been a rather complex soup. I have mentioned this in the manuscript more than once (I have cited the paper of John Sutherland’s group mentioned here [32], thanks). In line with the logic described here, even amino acids or peptides may have exist simultaneously with nucleotide and RNAs in the prebiotic environments, the idea of the RNA world as the beginning stage of life remains valid. The key point is that the establishment of RNA-peptide encoding mechanism is difficult from the very beginning. In other words, for example, when we say the emergence of an RNA replicase ribozyme in a nucleotide/RNA pool, we do not exclude the possibility of the simultaneous existence of amino acids/peptides in the same pool. Interestingly, according to a recent study, oligoarginine peptides may even take part in the replication of RNA by facilitating the dissociation of the product chain from the template chain of RNA [40]. However, they can yet merely be considered as environmental factors (just like water or ions), if only the “informatic mapping” (as another reviewer phrased it) from RNA to peptides did not exist. The lipid molecules, however, should have “explicitly” participate the development of the RNA world, by forming membranes which encompassed RNA molecules, giving rise to RNA-based protocell, as described in our concrete scenario regarding the RNA world.

In fact, the idea of the RNA world, as it was raised originally, is “biological” in concept. The word “RNA” therein refers to the central “life molecules” involved within the world. This “biological world”, if being placed into its “chemical” background, certainly was not only containing RNA molecules. Indeed, if we like, we can discern between a “biological” world and a “chemical world”— only a “chemical world” includes all the chemical components there around. In other words, when I depicted the scenario of the RNA world, I do not mean that other chemical components would not contribute to its origin and development.

Finally, as mentioned in the manuscript, it is just the studies from John Sutherland’s group that revealed a chemical process, possibly prebiotically reliable, to form activate monomers [23, 25, 26]. Indeed, the nucleotides produced this way are merely 2’, 3’- cyclic phosphates, which seem not highly active. But it is possible that these “semi-active” nucleotides may have been sufficient to launch the prebiotic assembly of RNAs, at least in the beginning of the RNA world.

Round 2

Reviewer 3 Report

The changes made in the manuscript do not modify significantly my opinion on the views of the author. This manuscript still represents a conventional approach to the RNA world hypothesis even though the author acknowledges that other components may have been present. However, in the response to my comments, the author states that these other biomolecules 'can merely be considered as environmental factors (just like water or ions)'. The problem with this views is that it makes the random emergence of a polymerase ribozyme so unlikely that it should still be considered as improbable even at the scale of the Universe since its origin. I claim that scientifc approaches to the origin of life must avoid hypotheses based on exceeding low probabilities (constituting actually a violation of the Second Law). The question of the origin of life is more related to the nature of the chemical processes that could have made the emergence of a polymer having the capacity of replicating through base pairing and some functional abilities less unlikely. No doubt that this process has required a far from equilibrium environment and complex (autocatalytic) reaction networks that probably predated some self-sustaining abilities of living organisms. Finding what kinds of contributions could mono- and oligo-nucleotides play in this network may help in understanding how the ability of ribonucleotides to reproduce their own sequence developed and thus allowed the system to evolve in an open-ended way. However, other entities (including amino acids, peptides and fatty acids) should not be considered as spectators but as essential players of the network as well as nucleotides. The chemistry of such processes is likely to be complex and will perhaps require decades of chemical investigation, but, in my opinion, origin of life scientists have to get rid of simplistic views as that of the RNA world hypothesis and of the dead end of an origin based on a single player. However, my opinion on the manuscript is more related to the pertinence of the RNA world hypothesis rather than to the specific work of the author, which represents an honest attempt to present the present state of that hypothesis even though it does not enough take into account its chemical relevance. I am therefore ready to reach the opinion of the other referees provided that extensive language checking is made (including the modified parts).